# Optimisation of Sample Preparation from Primary Mouse Tissue to Maintain RNA Integrity for Methods Examining Translational Control

**DOI:** 10.3390/cancers15153985

**Published:** 2023-08-05

**Authors:** June Munro, Sarah L. Gillen, Louise Mitchell, Sarah Laing, Saadia A. Karim, Curtis J. Rink, Joseph A. Waldron, Martin Bushell

**Affiliations:** 1Cancer Research UK Beatson Institute, Garscube Estate, Switchback Road, Bearsden, Glasgow G61 1BD, UK; 2School of Cancer Sciences, University of Glasgow, Garscube Estate, Switchback Road, Bearsden, Glasgow G61 1QH, UK

**Keywords:** translation, mRNA, ribosome, RNA integrity, polysome profiling

## Abstract

**Simple Summary:**

The rate by which the ribosome decodes an mRNA determines the final protein output of the mRNA. This can vary greatly between different mRNAs. Methods which allow accurate determination of translation rates of specific mRNAs require conditions that preserve native mRNA–ribosome interactions and thus cannot be denaturing-based. Complex tissues sometimes contain RNases that can breakdown the RNA. Here, we optimise methods in non-denaturing conditions enabling the application of methods that examine translational control to primary mouse tissue in an accurate and reproducible manner.

**Abstract:**

The protein output of different mRNAs can vary by two orders of magnitude; therefore, it is critical to understand the processes that control gene expression operating at the level of translation. Translatome-wide techniques, such as polysome profiling and ribosome profiling, are key methods for determining the translation rates occurring on specific mRNAs. These techniques are now widely used in cell lines; however, they are underutilised in tissues and cancer models. Ribonuclease (RNase) expression is often found to be higher in complex primary tissues in comparison to cell lines. Methods used to preserve RNA during lysis often use denaturing conditions, which need to be avoided when maintaining the interaction and position of the ribosome with the mRNA is required. Here, we detail the cell lysis conditions that produce high-quality RNA from several different tissues covering a range of endogenous RNase expression levels. We highlight the importance of RNA integrity for accurate determination of the global translation status of the cell as determined by polysome gradients and discuss key aspects to optimise for accurate assessment of the translatome from primary mouse tissue.

## 1. Introduction

A multitude of mechanisms have now been identified that act upon mRNAs to control both protein output and specifically where within the cell mRNA translation takes place [1,2,3,4]. It is now appreciated that mechanisms acting at this level can range from fine-tuning to dramatic changes in protein production. Overall, the amount of protein made from mRNAs expressed at similar levels can vary by two orders of magnitude, which demonstrates the level of gene regulation that translational control can exert [1].

Approaches that allow the analysis of the association of ribosomes with specific mRNAs are now being applied beyond cell lines into mouse models and patient samples [5,6,7,8,9]. These types of samples often offer additional challenges emerging from the higher and more variable levels of ribonucleases (RNases) present compared to those in cell lines. While there are RNases in the environment that we take measures to prevent getting into samples, there are also RNases naturally present within the samples that cannot so easily be avoided and are present at varying levels depending on the tissue. Although these RNases have important roles, such as in RNA turnover, in order to conduct various molecular biology investigations, their activity must be controlled during sample processing to minimise inappropriate activity from these endogenous enzymes during sample manipulation. RNases catalyse the breakdown of RNA; the two main classes are endoribonucleases and exoribonucleases. Exoribonucleases degrade the RNA from the terminal ends of the RNA molecules, and their activity can be classified as 5′→3′ or 3′→5′ [10]. Eukaryotic mature mRNA molecules have a 5′cap, a poly(A) tail, and can form a closed loop structure, all of which help to modulate RNA degradation by exoribonucleases [11]. Endoribonucleases, on the other hand, cleave RNA internally [12]. RNase A, an endoribonuclease, was first identified in the bovine pancreas [13,14,15], and various glycosylated forms have been identified—referred to as RNase B, C, and D [16,17]. Since then, other homologous RNases with structural conservation have been discovered, forming the RNase A superfamily that is unique to vertebrates. In this superfamily, there are eight canonical members present in humans (now named RNASE1-8) [18]; all of these are encoded on chromosome 14 [19]. Each of this superfamily of RNases contain a signal peptide that leads to them being secreted [20,21]. Residing extracellularly, they have roles in a range of biological processes, some of which are dependent on their ribonuclease activity, including dietary RNA digestion, clearance of circulating extracellular RNAs, immune regulation, antiviral activity in host defence, and angiogenesis [19,22,23,24,25]. Another family of endoribonucleases is the T2 family, distinguishing features of which are their optimal activity in more acidic environments compared to the RNase A family and different cleavage preference [26,27].

The activity of these RNases can be controlled endogenously by ribonuclease inhibitor (RNH1) [28,29]. This RNase inhibitor binds with strong affinity to members of the RNase A superfamily, preventing RNase activity by blocking the active site [30,31]. RNH1 localises to the cytosol, mitochondria, and nucleus where it protects intracellular RNA from degradation by these extracellular RNases [32,33,34]. During the processing of tissue samples, these protective mechanisms may be disrupted or overwhelmed, potentially resulting in increased mRNA degradation.

When working with RNA, high-quality RNA is integral to the quality of the data output. Methods examining the engagement of the ribosomes with mRNAs requires preservation of the mRNA–ribosome interactions and for the mRNAs to be fully intact to migrate through the gradient at the correct speed. Degraded RNA can lead to an altered gradient profile and thus inaccurate determination of the translational landscape. For example, an intact mRNA bound by six elongating ribosomes would migrate through a sucrose gradient at a certain density; however, if this mRNA were cleaved by an endoribonuclease, resulting in two mRNA fragments each bound by fewer ribosomes, these would each migrate at a reduced density. Reagents such as RNAlater provide a means to retain RNA integrity immediately at the point of tissue harvesting. However, a key mechanism by which these reagents act involves denaturing the proteins, which renders any RNases present non-functional but also other proteins within the sample. Translational profiling methods require non-denaturing conditions to preserve the integrity of the ribosome. In this paper, we utilise a range of alternative methods to preserve the RNA integrity of mouse tissue samples and demonstrate how this varies across different tissues. We also demonstrate how these improved methodologies can then be utilised to interrogate the translatome accurately and consistently in mouse tissues.

## 2. Materials and Methods

### 2.1. Animal Models

All experiments were performed in accordance with UK Home Office regulations and were approved by the University of Glasgow Animal Welfare and Ethical Review Board.

For the liver, colon, and small intestine tissue samples, female C57BL/6 mice aged 6–7 weeks purchased from Charles River were used. These experiments were carried out under the UK Home Office project licence PP6345023. The mice (5 per cage) were allowed access to food and water ad libitum and were kept in a 12 h day/night cycle starting at 7:00 until 19:00. Rooms were kept at 21 °C at 55% humidity. These animals were culled in accordance with Schedule 1.

For the lung and pancreas tissue, stock mice were housed in individually ventilated cages (IVCs) with access to standard irradiated diet and sterilised water pouches ad libitum. Mice were handled and assessed within sanitised laminar flow changing hoods. Some experimental mice were housed in conventional caging with access to diet and water ad libitum. Environmental enrichment was provided with nesting, fun tunnels, and wooden chews, with general cage maintenance being carried out by the Biological Services Unit at the Beatson Institute. These mice were bred and maintained in house. 

Work for the lung tissue samples was conducted under project licence number PE47BC0BF. Mice were culled with rising volume of CO_2_, with cervical dislocation as a secondary schedule 1. This was performed at 11 weeks for control lung tissue and at approximately 6 months post-induction with 1 × 10^7^ pfu SPC-Cre for tumour tissue (mice approximately 34 weeks of age). Tumour-bearing mice were of the genotype *Kras*^G12D/+^, *Myc*^−/+^, and iRFP+. Lungs were flushed with PBS prior to sampling to remove residual red blood cells and then moved to a sterile surface sprayed with RNaseZAP (Sigma; Cat: R2020) for dissecting out individual tumours. Both male and female mice were used, and when the amount of sample was limiting, samples were pooled to provide sufficient material for polysome gradient analysis.

Work for the pancreas tissue was conducted under Project licence PP8411096. Following weaning, mice were ear notched and the tissue sent to Transnetyx (Cordova, TN, USA) for DNA genotyping via real-time PCR to detect allelic alterations in genes of interest, including *KRAS* and *Trp53*. Female wild-type mice of approximately 100 days of age, bred on a mixed background, were used throughout these experiments. Following euthanasia, 70% ethanol was applied to the abdomen of the mice and the peritoneum was opened to inspect the internal organs. When collecting these pancreas tissue samples, dissection instruments were cleaned and sprayed with RNaseZAP (Sigma; Cat: R2020) before removing small pieces of tissue (~30 mg), which were placed in RNAlater (Thermo Fisher Scientific, Waltham, MA, USA; Cat: AM7021) or DNA/RNA Shield (Zymo Research; Cat: R1100-8-S) and stored at −80 °C until further use.

### 2.2. Tissue Preparation

Tissue pieces were flash frozen in liquid nitrogen and stored at −80 °C until required. The CryoGrinder system (OPS Diagnostics, Lebanon, NJ, USA), which ensures the samples remain frozen on nitrogen, was then used to grind the tissue with a mortar and pestle. To aliquot tissue in a manner that avoided the risk of thawing, empty 1.5 mL tubes were pre-marked, by comparing to a 1.5 mL tube with water in it; marks were made at 20 µL (for tests) or 200 µL (for sucrose gradients). Marked tubes were then chilled and kept on dry ice throughout dispensing—tipping the ground tissue up to the mark, which was performed in a cold room. Tissue aliquots and stocks were stored at −80 °C until required.

### 2.3. Cell Lysis

Lysis buffer was made fresh each time and comprised 1× gradient buffer (150 mM NaCl, 15 mM MgCl_2_, 15 mM Tris-HCl pH 7.5, 100 µg/mL cycloheximide (the gradient buffer can be stored at −20 °C), 1% Triton X-100, 0.05% Tween-20, 0.5 mM DTT, 5 mM NaF, 1× cOmplete™, Mini, EDTA-free Protease Inhibitor Cocktail (Roche, Basel, Switzerland), 1× Protease Inhibitor Cocktail for plant cell and tissue extracts (Merck), 2% n-Dodecyl-beta-Maltoside Detergent (Thermo Scientific, Waltham, MA, USA). For lysis tests, 200 µL lysis buffer (with either 1000 U/mL Ribolock (Thermo Scientific), 1000 U/mL Superase.In (Invitrogen, Waltham, MA, USA), 1 mg/mL Heparin (Merck, Darmstadt, Germany), 1000 U/mL, Murine RNase Inhibitor (New England Biolabs, Ipswich, MA, USA), or 10 mM Ribonucleoside Vanadyl Complex (New England Biolabs) was aliquoted and chilled on ice. A total of 20 µL of ground tissue sample was then tipped straight from dry ice into the lysis buffer. Tubes were immediately inverted to disperse the tissue fragments. Lysates were incubated on ice for 5 min with occasional inversion, then centrifuged at 13,000 rpm, 4 °C, for 5 min. A total of 150 µL of supernatant was retained and 1 mL TRIzol added (Invitrogen).

### 2.4. Total RNA Extraction from TRIzol

TRIzol/lysate mixes were incubated at room temperature for 5 min. A total of 200 µL chloroform was added, and samples were shaken vigorously for 30 s, then spun at 4 °C, 12,000 rpm, for 20 min in an Eppendorf 5415R centrifuge. The aqueous phase (~550 µL) was transferred to a new 1.5 mL Eppendorf tube. To precipitate the RNA, 2 µL glycogen (Roche) and 500 µL isopropanol were added and inverted to mix, then incubated for a minimum of 45 min on ice. Samples were then pelleted via centrifugation at 4 °C, 13,000 rpm for 30 min. Pellets were washed twice in ice-cold 750 µL 75% ethanol and spun at 4 °C, 13,000 rpm for 5 min each time. After the final wash removal, pellets were air-dried for 2 min then resuspended in 15µL RNase-free water.

### 2.5. Agarose Gel Electrophoresis

Total RNA concentration was determined via nanodrop and 0.5–1 µg was mixed with loading dye (New England Biolabs) and resolved via electrophoresis on a 1% agarose 1× TAE (40 mM Tris, 20 mM Acetate and 1 mM EDTA) with SYBR™ Safe DNA Stain (Invitrogen, 1:1000 dilution) gel at 150 V for 20 min then visualised on a Biorad Chemidoc Imaging system (Blue imaging tray). One kb HyperLadder (Bioline) was loaded on each gel as a reference.

### 2.6. Sucrose Gradients

Cell lysis was performed on 200 µL of tissue in 900 µL lysis buffer as described above. Tissue was lysed on ice for 5 min, centrifuged at 13,000 rpm, 4 °C, for 5 min and 800 µL of lysate transferred to a fresh tube. To obtain a corresponding total lysate sample, 40 µL of the lysate was taken and 1 mL TRIzol added. As a quality control, in parallel, 40 µL of tissue was taken and 1 mL TRIzol added directly to the tissue. This allows for the assessment of the RNA integrity within both the whole tissue and total lysate used for the gradients. A total of 600 µL of lysate was then loaded onto 10–50% sucrose gradients. This corresponds to 183 µg and 151 µg RNA for the liver tissue, 17 µg and 24 µg for the control lung tissue, 42 µg and 82 µg for lung tumour tissue, 56 µg and 69 µg for small intestine samples without inhibitors and 70 µg and 105 µg for small intestine samples treated with inhibitors. Gradients were spun at 38,000 rpm for 2 h at 4 °C in a Beckman Optima XPN90 centrifuge. Gradients were fractionated and collected using a Biocomp Gradient Station with attached Gilson FC203B fraction collector. Of note, with this machine, the final fraction remains in the gradient tube and is not run through the fractionator—this sample is manually retained as an additional final fraction (fraction 12); hence, there is an apparent extra fraction in the agarose gel RNA extractions compared to the gradient traces. Gradient fractions were collected into 7.7 M Guanidine HCl (3 mL) with glycogen (8 µL) and vortexed, then 4 mL of 100% ethanol was added, invert mixed, and precipitated at −20 °C overnight.

### 2.7. RNA Extraction of Sucrose Gradient Fractions

Gradient fractions were spun in a Beckman 5810R bench top at 4000 rpm, 4 °C, for 1 h. All the supernatant was removed, and pellets were resuspended in 320 µL RNase-free water. A total of 300 µL was transferred to a fresh 1.5 mL Eppendorf and 2 µL glycogen, 30 µL 3 M sodium acetate pH 5.2, and 900 µL 100% ethanol were added. Samples were inverted to mix and precipitated overnight at −20 °C. Samples were pelleted via centrifugation for 40 min, at 13,000 rpm and 4 °C. Pellets were washed twice with 750 µL 75% ethanol and centrifuged each time at 13,000 rpm for 5 min at 4 °C. Pellets were air-dried for 2 min and resuspended each in 25µL RNase-free water. Equal volumes of each fraction were loaded onto a 1% agarose/1× TAE/1:1000 SYBR™ Safe gel (usually around 3 µL) to check RNA integrity across the fractions. Fraction numbers on the agarose gels reflect everything up to that number on the gradient trace *x*-axis, i.e., 0–1 on the gradient profile is denoted as fraction 1 on the agarose gel and 1–2 on the gradient profile is denoted as fraction 2 on the agarose gel. As detailed in the previous section, fraction 12 is the final fraction that remains in the gradient tube and is not run through the fractionator due to the nature of the machine used. RNA samples were stored at −80 °C.

### 2.8. RNA Integrity Value Calculations

For the agarose gels, RNA integrity values were calculated as the total signal of the 28S and 18S rRNA bands compared to the whole lane using Fiji software (version ImageJ2) [35]. Then, the RNA integrity of the different lysis conditions was calculated relative to directly adding TRIzol to the tissue. As an alternative method, the Agilent TapeStation was also used to obtain RNA integrity numbers (RIN^e^); the lower marker, 18S rRNA and 28 rRNA, are indicated on the traces. Of note, the RNA samples were run on a TapeStation after long term storage at −80 °C and following multiple freeze thaws, whereas the agarose gels were conducted close to the time of initial processing.

### 2.9. RNase Expression Level Data

Mouse RNA-seq data normalised as transcripts per million was obtained from Söllner et al., 2017 [36]. 

## 3. Results

### 3.1. RNase Expression across Tissues

To determine which RNases may be most problematic when working with different mouse tissues, the expression of RNases in a range of mouse tissue types was assessed using a publicly available RNA-seq dataset [36]. Figure 1A shows the combined expression level (transcripts per million (TPM)) of the RNase A superfamily (RNASE1, RNASE2A, RNASE2B, RAF1 (RNASE3), RNASE4, ANG (RNASE5), RNASE6, and RAE1 (RNASE7)) across a range of mouse tissues. This identified the pancreas as a distinct outlier that has drastically higher RNase A superfamily levels in comparison to the other tissues examined (Figure 1A); this is in line with previous observations that pancreas tissue is particularly susceptible to RNA degradation [37,38,39]. One replicate of the duodenum, a part of the small intestine, showed higher RNase levels; given its proximity to the pancreas, this variability may be due to differences in the precision of tissue harvesting. The liver also showed heightened expression of RNase A superfamily members compared to other tissues (Figure 1A). Looking at the RNases individually showed differences in the abundance of the transcripts encoding the different RNase A superfamily proteins (Appendix A), with *RNASE4* and *ANG* being the highest in liver (Appendix A), and *RNASE1* dominating in the pancreas (Figure 1B). Other RNases also showed variability across the tissues; for example, *RNASE2B* has low expression in most cases but is elevated in the oesophagus (Appendix A), and *RAF1* (*RNASE3*) is expressed broadly across tissues, yet only weakly in the pancreas (Appendix A). In addition, the expression levels of RNase T2 family members were highest in the kidney, thymus, and colon (Figure 1C,D).

Cells can also express an endogenous RNase inhibitor that localises to the cytosol to protect intracellular RNA molecules from degradation by these secreted RNases [28,29]. Figure 1E shows the expression level of *RNH1* across the tissues. Oesophagus tissue has the highest expression level of this endogenous inhibitor and, interestingly, *RNH1* is not highly expressed in the pancreas despite the high levels of RNase in pancreas tissue. Together, this analysis highlights that these different tissues have different types and amounts of RNases along with different levels of endogenous inhibitors, suggesting that differential processing of samples may be required to obtain intact RNA from these different tissues.

### 3.2. Retaining RNA Integrity in Non-Denaturing Conditions

As standard methods for harvesting tissue for RNA analysis, such as RNAlater, are denaturing, we sought to assess the range of methods of RNase inhibition available that would allow the preservation of RNA integrity in non-denaturing conditions and will therefore keep the ribosome intact and engaged with mRNAs. First, tissues were flash frozen in liquid nitrogen as rapidly as possibly to minimise the time for action of endogenous RNases during dissection and to preserve ribosome positioning upon tissue harvesting. To maximise tissue lysis efficiency, the tissues were then ground with a pestle and mortar using the CryoGrinder system to keep the tissue frozen on nitrogen during the grinding process. Grinding the tissue increases the surface area and hence increases lysis efficiency, which means that lysis times can be kept to a minimum, further reducing the potential impact of any endogenous RNases present in the tissue. The ground tissue was then aliquoted on dry ice for use in downstream experiments. As a positive control for maximum retention of RNA integrity, TRIzol reagent was added directly to ground tissue; the guanidium thiocyanate component is chaotropic and denatures proteins present, and hence RNases are rapidly inactivated upon lysis in TRIzol. As a negative control to reflect the natural impact of endogenous RNases in the tissue on RNA quality, tissue samples were harvested in lysis buffer without any RNase inhibitors.

In these tests, we added a range of different commonly available RNase inhibitors to the lysis buffer (Table 1). Heparin is a broad-spectrum, non-specific RNase inhibitor that has been readily used when conducting polysome gradients in cell lines as a means to preserve RNA integrity across the gradient [40,41]. Ribonucleoside vanadyl complex (vanadyl) is also non-specific and reversibly inhibits several RNases [42,43]. RiboLock is a recombinant protein sourced from an *E. coli* strain expressing a mammalian RNase inhibitor (Thermo Scientific). Similarly, murine RNase inhibitor is a recombinant murine protein expressed and purified from an *E. coli* strain (New England Biolabs). Both inhibitors specifically target RNase A/B/C in a non-covalent manner with high affinity, with B and C being glycosylated forms of RNase A [16,17]. Finally, we used SUPERase.In which, in addition to inhibiting RNases A/B/C, also targets RNase I and RNase T1 (Invitrogen).

For the lysis of the tissue, aliquoted ground tissue was kept on dry ice and tipped directly into a 1.5 mL tube containing ice-cold lysis buffer, inverted, and lysed on ice. After centrifugation, RNA was isolated from the lysate using TRIzol. RNA integrity was determined by running the extracted RNA on an agarose gel. Through this method, the 28S (4.7 kb) rRNA and 18S (1.9 kb) rRNA bands are clearly visible and provide a strong indicator of RNA integrity a combination of clearly defined rRNA bands, the relative ratio between 28S and 18S rRNA (~2.0), and no smearing or additional banding being indicative of high RNA integrity [44,45,46]. It has been noted that the 28S/18S ratio alone is not always sufficient to detect partial RNA degradation as it does not account for additional sub-peaks of partially degraded 28S and 18S [44]; considering the information across the full lane of the gel can be more informative, as is the case with RNA integrity number (RIN) calculations [45]. Therefore, RNA integrity values were calculated here as the total signal of the 28S and 18S rRNA bands compared to the whole lane using Fiji software (version ImageJ2), and we also compare these values to RIN^e^ values obtained using the TapeStation [35]. The aim in this study was to optimise lysis conditions that were non-denaturing and able to obtain RNA of the same quality as a denaturing based extraction, so the RNA integrity of the different lysis conditions was calculated relative to TRIzol added directly to the tissue as a positive control.

Here, we have focused on tissues that cover a range of RNase activity levels. Comparing the lysis of the tissues directly in TRIzol to lysis buffer with no RNase inhibitors present, we observe a clear reduction in RNA integrity across the tissues (Figure 2, Appendix A). The extent of this varies depending on the tissue—from most to least RNA degradation level: pancreas >> small intestine > lung > colon > liver. This demonstrates the need for optimisation of lysis conditions to retain RNA integrity in non-denaturing conditions.

First, the liver tissue was assessed and, despite the noted high expression of *RNASE4* and *ANG* (Appendix A), the liver consistently showed prominent 28S and 18S rRNA bands (Figure 2A, Appendix A). Although, even without RNases inhibitors, the liver-isolated RNA showed good integrity compared to direct denaturing lysis in TRIzol, the addition of RNases inhibitors did increase the reproducibility of this (Figure 2A, Appendix A). This suggests RNase inhibitor addition, even at lower concentrations, may help to provide optimal and reproducible RNA integrity in liver tissue. For both lung and colon, SUPERase.In did not greatly improve the RNA integrity as there was still substantial 28S rRNA degradation. Nevertheless, RiboLock, murine RNase inhibitor, and heparin all improved the RNA integrity in these tissues (Figure 2B,C, Appendix A). The small intestine was subject to higher levels of endogenous RNA degradation, with complete loss of 28S rRNA integrity and a smear of RNA on the agarose gel indicating poor RNA integrity in the absence of RNase inhibitors (Figure 2D). Unlike with the previous tissues, heparin was not as effective at improving RNA integrity in the small intestine; rather, the more specific inhibitors RiboLock and murine RNase inhibitor that focus on targeting the RNase A family and associated glycosylated variants were more effective (Figure 2D, Appendix A). Of note, across the tissues investigated, vanadyl had variable impact on the integrity of the RNA. Although there was no classical smearing degradation pattern, there was repeatedly a reduced intensity of the 28S and 18S rRNA signal suggestive of some level of negative impact on rRNA integrity or impact on the capacity for the RNA to be purified downstream. Overall, the murine RNase inhibitor and RiboLock were the most effective at retaining high RNA integrity across tissues.

### 3.3. The Complexity of the Pancreas

The pancreas has very high RNase levels (Figure 1A,B), making it a particularly difficult tissue to work with when studying RNA-based methods. Flash freezing of the pancreas tissue followed by grinding on nitrogen already led to major RNA degradation, even when extracting the tissue directly in TRIzol, and this was not improved by lysis in the presence of RNase inhibitor (Figure 2E, Appendix A). Poor RNA integrity was observed consistently from directly TRIzol-extracted pancreas tissue (Figure 2F), suggesting that the RNA was already degrading at the point of tissue harvesting in the mice. Protocols for RNA-seq from pancreas tissue often feature the tissue being harvesting and placed straight into RNAlater solution (Invitrogen) [33]. Figure 2G shows that immediate harvesting of the pancreas tissue into RNAlater can lead to improved RNA integrity within the pancreas, although the reproducibility of this is variable despite consistency in the protocol; this demonstrates that the RNA integrity should always be checked before downstream investigation. DNA/RNA shield is another reagent for RNA preservation, but we did not see improved RNA integrity using this (Figure 2H). These data highlight the importance of tissue harvesting in maintaining RNA integrity. This may limit the application of RNA-based methods that require non-denaturing conditions to preserve interactions between cellular components within tissue types such as the pancreas.

### 3.4. Polysome Profiling in Tissue

There have been a range of next-generation sequencing techniques developed to allow detailed, high-resolution studies of mRNA translation, in particular polysome profiling [47] and ribosome profiling [48]. The basis of these methodologies is the identification of which mRNAs are associated with ribosomes. Cycloheximide, a translation elongation inhibitor [49], is added to lysis buffer to immobilise ribosomes on the mRNA as rapidly as possible upon cell lysis. This is followed by sucrose density gradients used to sediment ribosomes via centrifugation, which separates mRNAs based on the number of ribosomes residing on the mRNA [47], (Figure 3A). Each ribosome contains the corresponding ribosomal RNA (rRNA), which can be detected by passing the sample through a UV detector and measuring absorbance at 254 nm. Assessing this across the sucrose density gradient creates a profile for a given sample of the mRNAs with different numbers of ribosomes located on them (Figure 3A). At the top of the gradient lies the free tRNA and ribonucleoprotein complexes, then the early fractions contain the 40S pre-initiation ribosomal subunit either without or in complex with the mRNA (43S and 48S, respectively), the large 60S ribosomal subunit or the 80S monosome (Figure 3A), followed by the polysomes, where each additional peak is indicative of an additional ribosome residing on the mRNA (Figure 3A). The polysome gradient profiles show an overview of mRNA translation occurring in the cell. The number of ribosomes residing on an mRNA is a strong indicator of mRNA translation rate—i.e., in general the more ribosomes on an mRNA, the more translated the mRNA and the greater the protein output. However, there are exceptions to this, for example, due to reduced elongation rates or ribosome stalling post-initiation [50,51]. The translation of a specific mRNA of interest can be investigated through extraction of RNA from each of the gradient fractions, followed by qPCR or Northern blot across these [47]. Next-generation sequencing now enables global quantification of polysome association by sequencing of the pooled sub-polysome and pooled polysomes fractions, as well as corresponding total RNA population [52,53,54].

Having determined that liver tissue has consistent RNA integrity in non-denaturing conditions (Figure 3B, Appendix A), we then generated polysome gradient profiles to look at the global translation in the liver (Figure 3C, Appendix A); this showed relatively high ribosome occupancy in the liver (Figure 3C, Appendix A). Polysome profiles provide a global view of the translational status of the tissue via measuring the UV 254 nm absorbance across the fractions of a gradient. However, degradation of the RNA can lead to dramatic changes in profiles and thus render such examination inaccurate. This is because an mRNA cleaved by RNases could become multiple smaller fragments occupied by fewer ribosomes that would migrate differently through the sucrose gradient and therefore not represent the complete mRNA and the full complement of ribosome residency along it. Hence, a typical gradient profile for degraded RNA has an increased 80S peak and reduced polysome-associated mRNA. Therefore, it is important to check the RNA integrity of the total RNA lysate and that of the individual gradient fractions. Running equal volumes of each extracted fraction on an agarose gel clearly shows the presence of the 40S ribosome containing pre-initiation complexes (fraction 2), then the presence of the 80S ribosome (fraction 3 and above) (Figure 3D, Appendix A).

Lung tissue showed consistent RNA integrity in the presence of RNase inhibitors including RiboLock and murine RNase inhibitor (Figure 2B, Appendix A), and when treated with these two inhibitors in combination (Appendix A). So, we opted to use this combination for the lysis to assess the global translation in the lung to ensure RNA integrity is maintained through the experimental procedure (Figure 4A–C, Appendix A). This showed a low polysome to sub-polysome ratio in the lung compared with the liver, suggestive of low protein output or differential rates of translation initiation versus elongation. Importantly, assessment of the RNA integrity across the gradient fractions (Figure 4C, Appendix A) demonstrates that the reduced profile of the lung is not due to RNA degradation, but a genuine result showing relatively fewer ribosomes within the polysome region of the gradient. To interrogate mRNA translation in the lung further, we also conducted sucrose gradients for lung tumour tissue (Figure 4D–F, Appendix A). Lung tumours were initiated by induction of *Myc*ER^T2^ and *Kras*^G12D^ and tumour tissue samples were collated 6 months post-induction to obtain large tumours [55,56]. This showed that the same lysis conditions were sufficient to maintain RNA integrity in the tumour tissue (Figure 4D, Appendix A) and across the corresponding gradient profiles (Figure 4E,F, Appendix A).

Given the variable RNA integrity of the small intestine, we next compared polysome profiles of small intestine lysed in the presence and absence of RNase inhibitor. To try to improve upon initial testing (Figure 2D, Appendix A), we also trialled a combination of two of the most effective inhibitors—RiboLock and murine RNase inhibitor (Appendix A)—and opted to use this combination for polysome gradient experiments to maximise RNA integrity retention throughout these experiments. For polysome profiling, it is also important to take a corresponding total RNA sample of the lysate loaded on to the gradient as a reference for use in downstream qPCR or sequencing. The small intestine lysed in the absence of inhibitors showed poor total RNA integrity with complete loss of the 28S rRNA, whereas the combination of RNase inhibitors greatly and reproducibly improved the total RNA integrity (Figure 5A, Appendix A). The corresponding polysome gradient profiles reflect this differential RNA integrity and are very distinctive, with a greatly reduced amount of RNA present in the polysomes and increased monosome and disome peaks in the absence of an RNase inhibitor (Figure 5B, Appendix A). This suggests that active RNases have cleaved mRNAs occupied by several ribosomes into multiple shorter fragments, each occupied by fewer ribosomes. Examination of the RNA integrity of the individual polysome gradient fractions in these two conditions clearly demonstrates that the reduced polysome occupancy in the no inhibitor condition is a result of large losses of RNA integrity across the gradient (Figure 5C,D, Appendix A). Overall, this demonstrates the importance of checking RNA integrity, particularly in non-denaturing conditions such as polysome profiling, and how misinterpretations can result if these critical quality controls are not conducted.

## 4. Discussion

For RNA-based techniques, RNA integrity is important for ensuring the quality and reproducibility of downstream results. The quality of the RNA is even more important when examining translational control mechanisms which use polysome profiles and other methods for determining the association of ribosomes with mRNAs [35,40]. Here we have demonstrated key aspects in the handling and processing of tissue samples that impact on RNA integrity and how, from the set of tissues tested in this study, some may be particularly susceptible to degradation due to their endogenous RNASE1 levels.

The RNase inhibitors used in this study had differing degrees of success in improving RNA integrity. SUPERase.In only slightly improved integrity, while heparin improved integrity of the liver, lung, and colon, but was not sufficient in the small intestine. RiboLock and the murine RNase inhibitor led to the greatest and most consistent improvements in RNA integrity across the tissues tested in this study (Figure 2, Appendix A). SUPERase.In, RiboLock, and murine RNase inhibitor all have the capacity to inhibit the RNase A family and were used at the same working concentration of 1000 U/mL (Table 1), yet SUPERase.In was unexpectedly not as effective at inhibiting the RNases in our given conditions (Figure 2, Appendix A). This may be due to differences in how ‘units’ are defined between companies, or, as RiboLock and murine RNase inhibitor specifically target only the RNase A superfamily and the glycosylated forms, perhaps targeted inhibition of specific RNases is more effective than more broad-spectrum inhibition in these conditions and tissue types. Heparin and vanadyl are more cost-effective per sample (Table 1) and, in the case of liver, are equally suitable; however, it is likely that, in tissues such as the liver, there is room for reducing the concentration of the other inhibitors and hence decreasing the cost per sample. Furthermore, heparin is problematic in downstream applications as it inhibits reverse transcription reactions, so it must be removed using LiCl precipitation for additional investigation of the samples [57]. Overall, it depends on the exact application and spectrum of RNases that are encountered as to which may be the most appropriate inhibitor for a given experiment. In this context, RiboLock and the murine RNase inhibitor were found to be the most effective inhibitors in aiding the preservation of RNA integrity and ribosome–mRNA interactions in these tissues.

It is aIso important to note that differences in tissue sample handling may be a key source of variability in RNA integrity. This can be an unpredictable source of variability as often when collecting samples from mice the person harvesting the tissue each time may differ, or the person processing the samples downstream may be different. This emphasises the importance of checking the RNA integrity of all samples and how including best practices from the very start of the sample handling may help improve consistency.

Having determined optimal lysis conditions for high-quality RNA, we have demonstrated how these can be utilised to accurately examine the association of mRNA with ribosomes in liver, lung, and small intestine and how, without this key quality control, results could be misinterpreted. Polysome gradient profiles demonstrated how different tissues have very different levels of ribosome occupancy, highlighting the importance of being able to accurately study translation in a wide number of tissues.

As for the pancreas, we show the difficulties in conducting RNA-based methodologies with this tissue. None of the non-denaturing based lysis methods tested in this study were sufficient to preserve both RNA and ribosome integrity (Figure 2E, Appendix A); thus, study of mRNA translation in pancreas tissue requires greater optimisation, and perhaps new or combinations of methods of RNase inhibition. Importantly, this study also highlights the extreme extent of RNases in the pancreas (Figure 1A,B) and the importance of sample handling from the very start of the harvesting procedure for the preservation of RNA integrity (Figure 2F–H). These findings will help improve RNA integrity in pancreas tissue harvested, even for standard RNA-seq. Additional methods have recently been explored that include modifications that might be useful for making further improvements. For example, direct injection of denaturing RNAlater solution into the pancreas immediately after extraction [38,58] or, most recently, perfusion of RNAlater prior to excision have shown promise in improved pancreas RNA integrity [59]. In the future, direct perfusion of non-denaturing inhibitors such as murine RNase inhibitor may lead to a reliable way to harvest pancreas tissue for experiments that require non-denaturing conditions for downstream procedures. Further study is required to determine if these additional steps alter ribosome occupancy and whether extra steps, like combining with cycloheximide infusion, are needed to preserve the endogenous translational landscape.

Ribosome profiling is another commonly used technique to study translational control, which differs from polysome profiling as it allows codon resolution of the ribosomes [40]. This is commonly achieved with RNase I digestion, although other RNases have also been used [60], and utilises the fact that the ribosome protects from degradation the portion of the RNA on which it resides [48]. The use of RNase I as a tool provides an additional variable to account for, as the RNase inhibitor must not target RNase I (or an alternative RNase used) but inhibit the dominant RNases in the tissue, such as those of the RNase A family. In addition, RNase I is utilised due to its efficient cleavage at all four nucleotides, which prevents the introduction of bias into precise ribosome residency determination to nucleotide resolution. It is preferable for this methodology to be applied in this controlled manner as opposed to allowing the action of endogenous RNases that can be unpredictable and have known bias—RNase A cleaves at pyrimidines [61] or RNase T2 that cleaves at AU and GU dinucleotides [27]. Also, an important aspect of the corresponding downstream analysis when conducting ribosome profiling is the comparison to the overall mRNA expression level changes. This requires a total mRNA sample for which an undigested RNA sample is required. Here, we show that both murine RNase inhibitor and RiboLock are inhibitors which fulfil these requirements and that can be utilised to improve RNA integrity with non-denaturing tissue lysis.

This study highlights the importance of checking RNA integrity of all tissues and all replicates due to the variability of RNase-mediated degradation, even when endeavouring to keep the tissue harvesting protocol as consistent as possible. There are a range of methods that could be applied to check the RNA integrity, including running the RNA on an agarose gel, TapeStation, or bioanalyzer or looking more closely at the integrity of specific mRNAs using methods such as Northern blotting. This will ensure quality results and meaningful interpretation. In addition, it is probable these optimisations and inhibitor combinations could also be readily applied to other RNase-rich tissues, such as the mouse spleen, and to human tissue.

## 5. Conclusions

Overall, this study highlights the fundamental importance of RNA integrity for accuracy of downstream investigations and provides methods that consistently maintain RNA integrity in a range of different mouse tissues. These can now be utilised for accurate investigation of tissue samples with methodologies that require non-denaturing conditions.

## Figures and Tables

**Figure 1 cancers-15-03985-f001:**
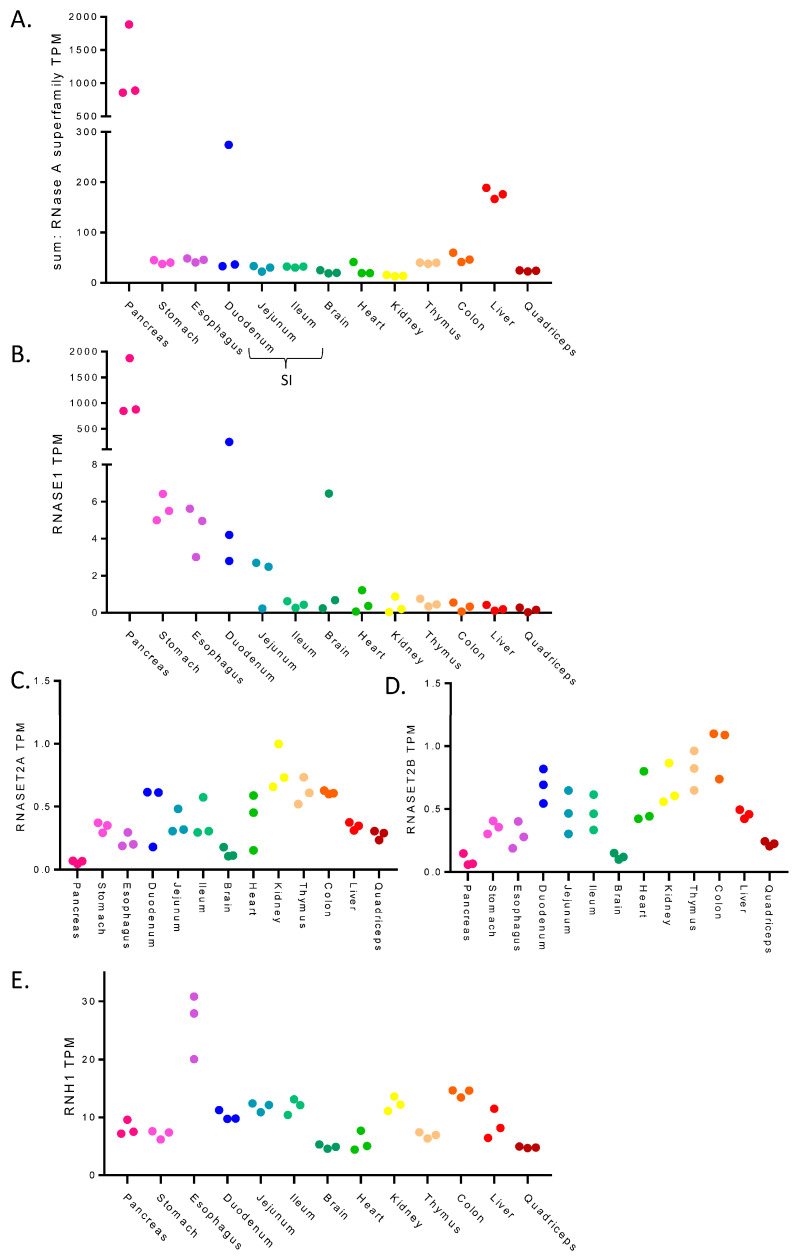
**Expression levels of RNase A family and RNase inhibitors across mouse tissues.** Expression levels (TPM: transcripts per million) of RNases and RNase inhibitors in several primary mouse tissues as determined using RNA-seq from Söllner et al., 2017 [36]. Each data point represents a replicate. (**A**) the total sum of RNA expression of RNase A superfamily members. (**B**) RNASE1 expression. (**C**,**D**) RNASET2 family members expression. (**E**) RNH1, an endogenous RNase inhibitor.

**Figure 2 cancers-15-03985-f002:**
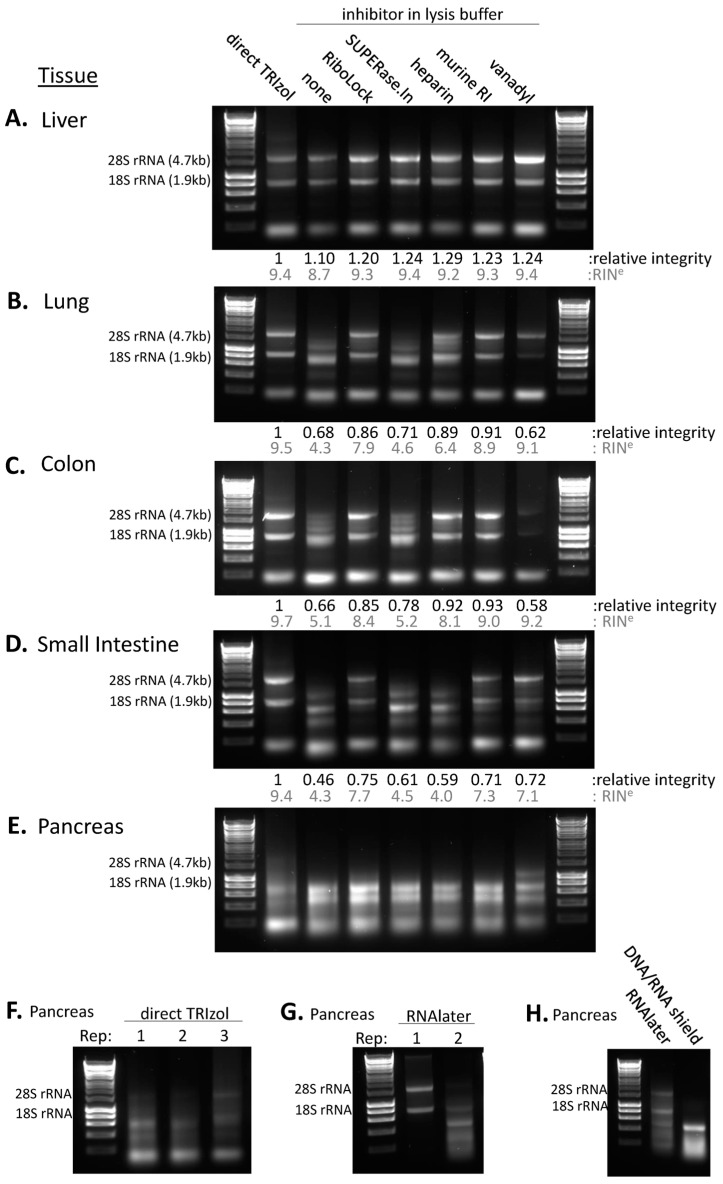
**RNA integrity across tissues under different non-denaturing lysis conditions.** (**A**–**E**) RNA integrity for primary mouse tissues lysed in non-denaturing buffer containing one of a range of RNase inhibitors. Tissue harvested directly in TRIzol was used as a positive control for relative RNA integrity measures shown. RIN^e^ values (grey) were obtained using the TapeStation. Ladder used is 1 kb HyperLadder. 28S rRNA and 18S rRNA sizes and expected positions are indicated. (**A**) Liver. (**B**) Lung. (**C**) Colon. (**D**) Small intestine. (**E**) Pancreas. (**F**) Three biological replicates of mouse pancreas tissue flash frozen in nitrogen and lysed directly in TRIzol. (**G**) Two biological replicates of mouse pancreas tissue place into RNAlater solution at the point of harvesting and then lysed directly in TRIzol. (**H**) Pancreas tissue placed in RNAlater solution or DNA/RNA shield solution at the point of harvesting and then lysed directly in TRIzol. Original western blots are presented in Appendix A.

**Figure 3 cancers-15-03985-f003:**
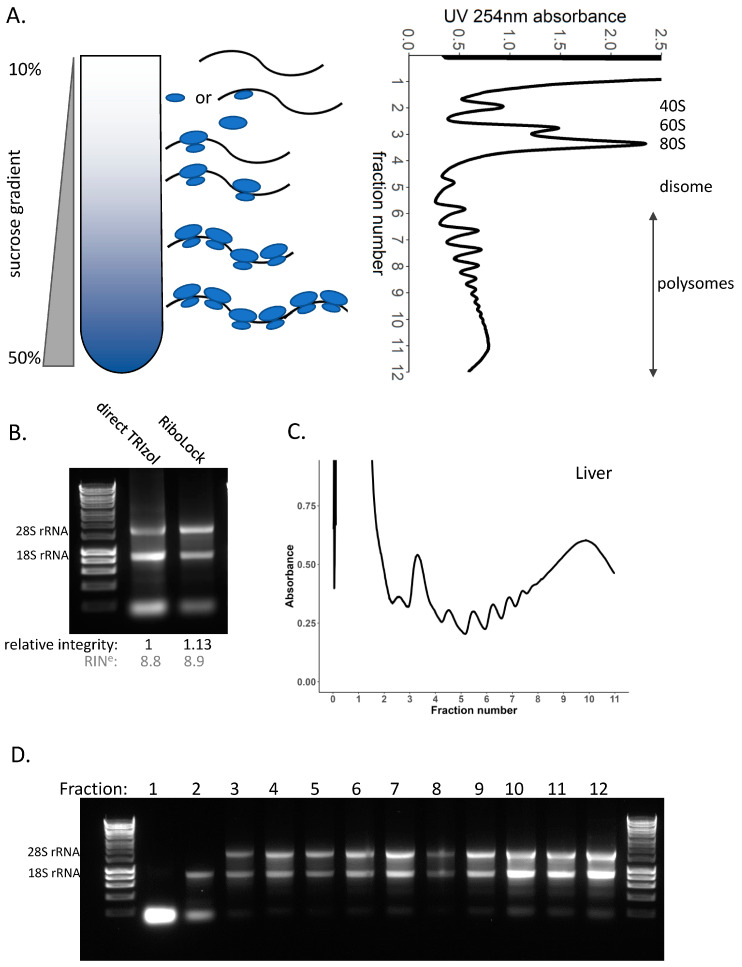
**Sucrose density gradients in liver tissue.** (**A**) Sucrose density gradients contain a range of sucrose concentrations from low to high concentration (in this case 10% to 50%). Lysate is loaded on top of the gradient and spun to sediment mRNAs based on the number of ribosomes bound. RNA detection at UV 254 nm across the gradient produces a trace—known as a polysome profile. This example is from a HEK293 cell line. (**B**) RNA integrity of liver tissue lysed directly in TRIzol and the total lysate lysed in non-denaturing lysis buffer containing RiboLock, loaded on to the gradient. RIN^e^ values (grey) were obtained using the TapeStation, corresponding traces are shown in Appendix A. (**C**) Polysome profile for mouse liver tissue. (**D**) RNA integrity across each of the individual gradient fractions. Original western blots are presented in Appendix A.

**Figure 4 cancers-15-03985-f004:**
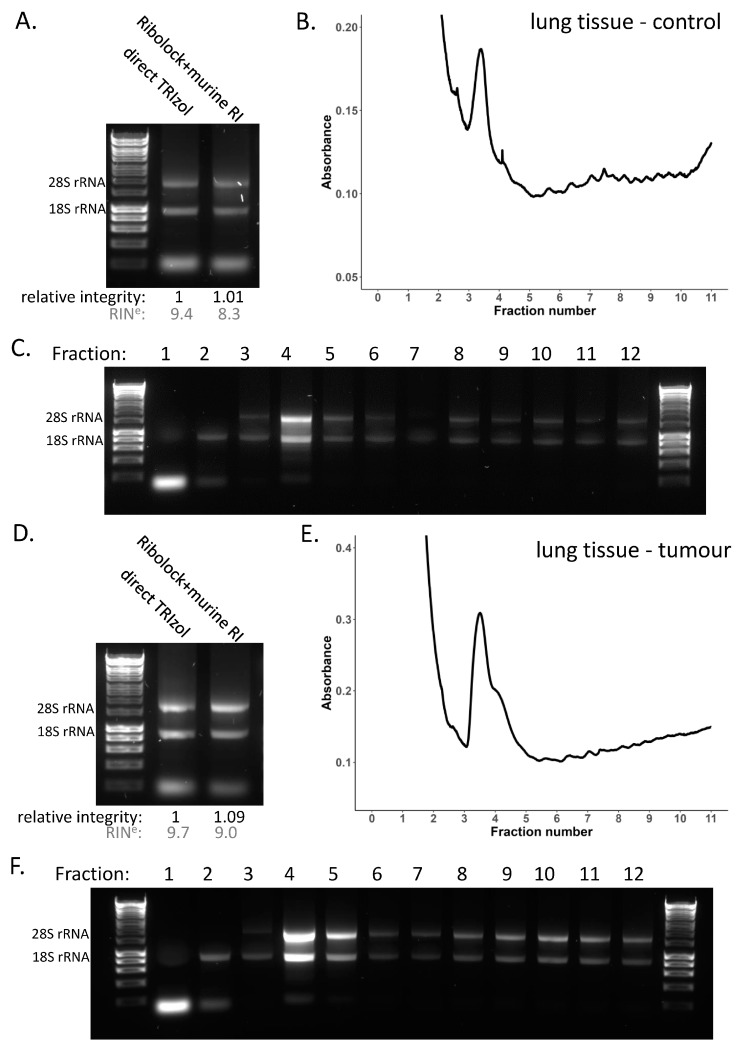
**Sucrose density gradients in lung tissue.** (**A**–**C**) Polysome profile in mouse lung tissue. (**A**) RNA integrity of lung tissue lysed directly in TRIzol and the total lysate, lysed in non-denaturing lysis buffer containing RiboLock, loaded on to the gradient. RIN^e^ values (grey) were obtained using the TapeStation, corresponding traces are shown in Appendix A. (**B**) Polysome profile for healthy mouse lung tissue as a control. (**C**) RNA integrity across each of the individual gradient fractions. (**D**–**F**) Polysome profile in mouse lung tumour tissue. (**D**) RNA integrity of lung tumour tissue lysed directly in TRIzol and the total lysate, lysed in non-denaturing lysis buffer containing RiboLock, loaded on to the gradient. RIN^e^ values (grey) were obtained using the TapeStation, corresponding traces are shown in Appendix A. Polysome profile for mouse lung tumour tissue. (**F**) RNA integrity across each of the individual gradient fractions. Original western blots are presented in Appendix A.

**Figure 5 cancers-15-03985-f005:**
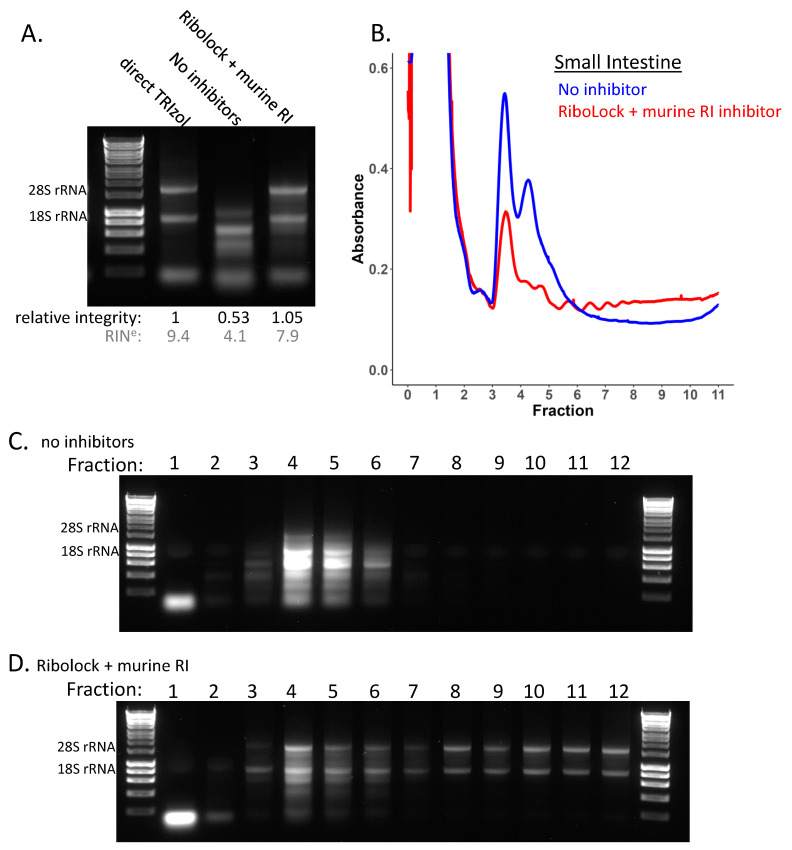
**Sucrose density gradients in small intestine tissue.** Polysome profiles of mouse small intestine tissue in the presence and absence of RNase inhibitor. (**A**) RNA integrity of small intestine tissue lysed directly in TRIzol and the total lysate, lysed in non-denaturing lysis buffer without RNase inhibitor or containing a combination of RiboLock and murine RNase inhibited loaded on to the gradient. RIN^e^ values (grey) were obtained using the TapeStation, corresponding traces are shown in Appendix A. (**B**) Polysome profile for mouse small intestine tissue, in the presence (red) and absence of (blue) RNase inhibitor. (**C**) RNA integrity across each of the individual gradient fractions for small intestine lysed in the absence of RNase inhibitors. (**D**) RNA integrity across each of the individual gradient fractions for small intestine lysed in the presence of a combination of RNase inhibitors RiboLock and murine RNase inhibitor. Original western blots are presented in Appendix A.

**Table 1 cancers-15-03985-t001:** **Comparison of commercially available RNase inhibitors used in this study.** Summary of RNase inhibitors used in this study, including supplier, current costings (correct as of June 2023), the working concentrations used in this study, target RNase families and some of the possible limitations.

Inhibitor	Supplier	Cost	Working Concentration	Targets	Cons
RiboLock	Thermo Scientific (Waltham, MA, USA)	£19.52/1000 U	1000 U/mL	RNase A, B & C	Limited targeting
SUPERase.In	Invitrogen (Waltham, MA, USA)	£68/1000 U	1000 U/mL	RNase A, B & C RNase T1 RNase I	Inhibition of RNases used in ribosome profiling (e.g., RNase I) limits its application
Heparin	Sigma-Aldrich (St. Louis, MO, USA)	£0.77/mg	1 mg/mL	broad spectrum	Can block downstream applications—requires LiCl precipitation to remove Inhibition of RNases used in ribosome profiling (e.g., RNase I) limits its application
murine RNase inhibitor	New England Biolabs (Ipswich, MA, USA)	£25.60/1000 U	1000 U/mL	RNase A, B & C	Limited targeting
Ibonucleoside Vanadyl Complex	New England Biolabs (Ipswich, MA, USA)	£4.10/10 mM	10 mM	broad spectrum	Need to add EGTA to remove after use Inhibition of RNases used in ribosome profiling (e.g., RNase I) limits its application

## Data Availability

A publicly available RNA-seq dataset was used in this study. This dataset is available at ArrayExpress: E-MTAB-6081.

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
