# Peer review of "Optimisation of Sample Preparation from Primary Mouse Tissue to Maintain RNA Integrity for Methods Examining Translational Control"

_cancers, 2023, doi:10.3390/cancers15153985_

Round 1

Reviewer 1 Report

The topic of this manuscript is very important for those researchers who investigate or are planning to investigate translation in primary tissue samples, and perhaps – also in some particular cultured cell lines with high RNAse activity. Especially important is the issue with sucrose gradients and polysome profiles, as the absence of polysomes can be either interpreted as repressed translation or as RNA degradation. Here, comparison of RNA integrity after cell lysis with direct Trizol isolation is a very good suggestion. In my opinion this is a very important manuscript and I will be happy to recommend it for publication.

Minor suggestions – it will be useful to include the working concentrations of inhibitors in Table 1.

Author Response

Reviewer 1 comments:

The topic of this manuscript is very important for those researchers who investigate or are planning to investigate translation in primary tissue samples, and perhaps – also in some particular cultured cell lines with high RNAse activity. Especially important is the issue with sucrose gradients and polysome profiles, as the absence of polysomes can be either interpreted as repressed translation or as RNA degradation. Here, comparison of RNA integrity after cell lysis with direct Trizol isolation is a very good suggestion. In my opinion this is a very important manuscript and I will be happy to recommend it for publication.

Minor suggestions – it will be useful to include the working concentrations of inhibitors in Table 1.

Response:

Thank you for your positive feedback. We have now added this information to the table. 

Reviewer 3 Report

This study provides very useful information, both in terms of the experiments conducted but equally the description of challenges, biological context, methodological choices etc. This, in its combination, would be quite difficult to find elsewhere especially for beginners in this field of experimentation.

To further improve it, I thought it would be good not to entirely rely on RIN number and visual agarose gel inspection for (m)RNA integrity assessment. In the 'old day' one would have run a Northern blot to confirm full-length mRNA for an exemplary mRNA. Can this be done? Or perhaps direct RNA sequencing with ONT could be a modern-day equivalent?

I found a few issues with grammar that could easily be corrected with another careful read.

Author Response

Reviewer 3: This study provides very useful information, both in terms of the experiments conducted but equally the description of challenges, biological context, methodological choices etc. This, in its combination, would be quite difficult to find elsewhere especially for beginners in this field of experimentation.

To further improve it, I thought it would be good not to entirely rely on RIN number and visual agarose gel inspection for (m)RNA integrity assessment. In the 'old day' one would have run a Northern blot to confirm full-length mRNA for an exemplary mRNA. Can this be done? Or perhaps direct RNA sequencing with ONT could be a modern-day equivalent?

Response: Thank you for your suggestion. Due to the tight time constraints imposed by the journal to respond to the reviewers comments, we have not been able to conduct Northern blot experiments, however we have added to the discussion regarding this. In addition, reviewer 2 comment 1 suggested a different method for determining the RNA integrity so, as suggested by reviewer 2, we have now included RINe values obtained via the TapeStation where possible.

Reviewer 4 Report

The authors have provided very useful information about the method to prepare lysates containing RNAs with high quality from tissues under non-denaturing conditions. I totally agree with the concept of this paper, which mentions the importance of RNA integrity/ quality in the samples, particularly for the downstream RNA analysis. I cannot judge whether this study fits well in this journal as this is the ‘cancer’ journal and therefore I leave this to the editors, but I find this study very informative and interesting. I have some minor comments below.

1) I like the data of Figure 2 comparing RNase inhibitors and am a little surprised that, although SUPERaseIn also inhibits RNase A, B & C, this inhibitor is not so effective compared to RiboLock and murine RNase inhibitor. Is it due to the differences in activity? Any discussion or suggestion would be helpful.

2) Figure 3C and D: fraction numbering is a little confusing, e.g. fraction 1 in Figure 3D corresponds to the fraction between 0 and 1 in Figure 3C? The author should amend all of the numbering in the sucrose density gradient data.

3) Line 364: ‘Hence, a typical gradient profile for degraded RNA has an increased 80S peak and reduced polysome-associated mRNA.’

…In Figure 5B, two prominent peaks between fractions 3 and 5 were increased in the lysates prepared without RNase inhibitors. I wonder which peak contains 80S and what the other peak is, and also how to assign the ribosomal subunit, 80S or disome?

4) Related with the comment above, in the NAR paper (PMID: 27638886), they detected the peak of ribosomes in the lysates prepared from mouse pancreas without heparin. However, according to the data here described in Figure 2E, rRNAs are degraded even using heparin. 80S peak with rRNAs degraded is also observed in Figure 5. I am curious about why the 80S ribosome, as detected in the gradient, remains even though rRNAs are degraded and also curious about whether this monosome with rRNAs degraded retains ‘protected’ mRNA fragments and can be used for ribosome profiling? It would be helpful if the authors can discuss this.

5) Also in the NAR paper, they detected no ribosomes from mouse spleen, but detected the ribosomes in the lysates prepared with heparin. I understand that this might be beyond the scope of this study but I wonder whether the authors can recover rRNAs in the lysates prepared from spleen using their method or not?

6) In the method section, there are typos such as ‘CO2’, ‘-20’C’, ‘Triton X100’, ‘Tween’, ‘ug’ etc…

7) It wound be helpful if the authors can provide the amount of RNAs, in the 600 microL of lysate, loaded onto the sucrose gradients.
